

# Feeding response of the tropical copepod *Acartia erythraea* to short-term thermal stress: more animal-derived food was consumed

Simin Hu[1], Sheng Liu[1], Lingli Wang[1,2], Tao Li[1,3] and Hui Huang[1,3]

[1] Key Laboratory of Tropical Marine Bio-resources and Ecology, Guangdong Provincial Key Laboratory of Applied Marine Biology, South China Sea Institute of Oceanology, Chinese Academy of Sciences, Guangzhou, Guangdong, China
[2] University of Chinese Academy of Sciences, Beijing, China
[3] Tropical Marine Biological Research Station in Hainan, Chinese Academy of Sciences, Sanya, Hainan, China

Corresponding author
Sheng Liu, shliu@scsio.ac.cn

## ABSTRACT

The objective of this study was to explore the feeding response of tropical copepods to short-term thermal shock and provide insight into the potential impact of coastal power plants on the trophic dynamics of tropical coastal ecosystems. Feeding experiments were conducted at three different temperatures (29 °C, 33 °C, and 35 °C) using the copepod *Acartia erythraea*, collected from Sanya Bay, China. The grazing rate of *A. erythraea* decreased dramatically in the high temperature treatment. Analysis of 18S rDNA clone libraries revealed that the diet of copepods from different treatments was mainly comprised of diatoms, metazoans, and protozoans; *A. erythraea* exhibited an obvious feeding preference shift with temperature, with a change from a diatom-dominated diet at 29 °C to a metazoan-dominated diet at 35 °C, and the omnivory index shifted from 0.1 to 2.84 correspondingly. Furthermore, *A. erythraea* showed a positive feeding response to plant food (i.e., phytoplankton and land plants) in the control treatment (29 °C), but a positive response to animal prey (i.e., metazoans and protozoans) at temperatures exceeding 33 °C, as evaluated by the Ivlev's selectivity index. Our results suggest that copepods could regulate their food intake by considering their energy demands when exposed to short-term thermal stress, which might influence the pathway of materials moving up the trophic system. However, further studies are required to elucidate the effects of elevated temperature on feeding of different organisms in order to predict the influence of thermal pollution on the food web of tropical coastal ecosystems.

## INTRODUCTION

Demand for electrical energy generation is increasing in coastal areas due to growing urbanization. Seawater-cooled power plants provide an economical way to generate electricity and growing numbers of power plants are operating in coastal areas, especially in subtropical and tropical regions (*Poornima et al., 2006*). The thermal effluent water discharged from the cooling systems of coastal power plants can cause
various environmental changes, with temperature elevation being the main impact on coastal habitats (*Jiang et al., 2009*); organisms are subjected to acute thermal shock with a temperature increase of approximately 6–10 °C above ambient temperature in tropical zones (*Poornima et al., 2006*; *Jiang et al., 2009*). Temperature increases can alter physiological processes and metabolic rates at the species level, and biomass and distribution at the community scale (*Sommer et al., 2012*; *Lewandowska et al., 2014a*). Furthermore, asynchronous responses by individual species to variations in temperature can further disrupt their interactions with other species and lead to mismatches in marine food webs (*Lewandowska et al., 2014b*). Thus, information on variations in trophic interactions under thermal stress is vital for understanding the functioning of coastal ecosystems, especially in tropical ecosystems where organisms generally live at ambient temperatures that are relatively close to their upper thermal limits (*Hoegh-Guldberg, 1999*; *Worthington et al., 2015*).

Planktonic organisms are usually small, with short generation times and weak motility. These characteristics make them especially vulnerable to sea temperature increases (*Hays, Richardson & Robinson, 2005*). Among the marine plankton, copepods represent the major group of secondary producers and play a key role in the cycling of nutrients and energy in marine ecosystems by forming a trophodynamic link from primary production to higher trophic levels (*Colombo-Hixson et al., 2013*). Understanding the feeding responses of copepods to thermal stress could provide insight into their survival and also the dynamics of coastal food webs. Copepods commonly inhabit the upper ocean layers (0–100 m), where they are exposed directly to temperature increases due to thermal effluents. These increases in temperature are fast and dramatic; therefore, short-term responses to thermal stimulation are important for species survival and adaptation, especially for thermally sensitive species (*Werbrouck et al., 2016*).

Feeding is an important basic process for organisms and is influenced by temperature both directly and indirectly (*Sommer et al., 2012*). Water temperature increases could affect the feeding process of copepods indirectly by exerting effects on food availability. Cell size, photosynthetic rates, and nutritional status (e.g., lipid composition, degree of fatty acid saturation, etc.) of phytoplankton are all affected by increased temperature (*Rousch, Bingham & Sommer, 2003*; *Zhang et al., 2012*). These changes in the nutritional status of food items might influence their palatability (*Peter & Sommer, 2012*; *Lewandowska et al., 2014b*). High temperatures can also affect physiological traits (e.g., body size, life cycle length, respiration, metabolic processes) of copepods directly (*Shiah et al., 2006*; *Jiang et al., 2008*; *Worthington et al., 2015*; *Dziuba, Cerbin & Wejnerowski, 2017*). Temperature increases of 3 °C have been shown to promote the growth and metabolic rate of copepods near power plants, especially in winter (*Wojtal-Frankiewicz, 2012*). However, slightly higher mortality of copepods was observed in the outlet regions of power plants than in the inlet regions, and copepod species with larger body sizes were more susceptible to thermal stress than smaller ones (*Jiang et al., 2008*; *Jiang et al., 2009*). Changes in copepod community structure could occur due to species-specific thermal tolerances and favor the dominance of copepods with smaller body sizes in areas near power plants. Elevated temperatures could

also reduce the swimming ability of copepods and affect their feeding behavior (*Larsen, Madsen & Riisgård, 2008*). Copepods can adjust their feeding strategy according to food availability; it has been reported that copepods are able to efficiently adjust their digestive enzyme pattern to a new food source within days (*Kreibich et al., 2011*). Copepods switched from feeding on phytoplankton to grazing on ciliates under warming conditions and exhibited stronger top-down control over protozooplankton under increased temperatures because protozooplankton had faster growth rates (*Lewandowska et al., 2014a*; *Aberle et al., 2015*).

Most studies in this field have been mesocosm experiments designed to predict the effects of global warming on the productivity of copepod communities in mid- and high-latitude areas/temperate seas (*Kjellerup et al., 2012*). Copepods in tropical nearshore waters are relatively close to their upper thermal limit, so any further increase in the ambient water temperature may severely affect their feeding behavior (*Chew & Chong, 2016*). However, little is known about the feeding responses of copepods exposed to thermal stress in tropical areas.

*Acartia* is a genus of small copepods that dominate zooplankton communities in temperate and subtropical coastal ecosystems. *Acartia* spp. usually lack storage capabilities and are vulnerable to environmental changes that require rapid responses to changing food availability (*Shin et al., 2003*). *Acartia erythraea* was selected for this study because it is an important coastal copepod in the South China Sea (*Liu et al., 2010*). *Acartia erythraea* is one of the dominant copepod species in subtropical (e.g., Daya Bay) and tropical (e.g., Sanya Bay) waters in summer and fall (*Hu et al., 2012*). Previous studies have shown that elevated temperatures clearly affect the feeding behavior of *Acartia* (*Henriksen et al., 2012*); however, their response to extreme thermal stress remains unclear. In the present study, a short-term feeding study on *A. erythraea* was conducted using the natural plankton community as a food resource under different temperature treatments, in order to investigate the rapid feeding response of copepods to thermal stress.

## MATERIALS AND METHODS

### Copepod collection and gut clearance

Zooplankton were collected at dusk on April 23, 2014, using surface plankton net-tows (diameter 50 cm, mesh size 505 $\mu$m) in the coastal waters of Luhuitou, Sanya Bay (109°28′E, 18°12′N). Live copepods were kept in a 1-L plastic container and transferred to the laboratory for gut evacuation. For the feeding experiment, surface seawater was also collected using a clean 25-L bucket in the same region.

Copepods were immediately transferred into several 500-mL beakers containing 0.45 $\mu$m-filtered seawater for gut evacuation that was performed in the dark for >48 h. Every 4 h during the clearance period, the copepods were gently collected using 200-$\mu$m mesh and rinsed 3–5 times with fresh seawater (FSW) to remove any attached detritus, and then transferred to FSW to fully empty their gut contents. After this procedure, healthy adult *A. erythraea* females of similar size were selected for the feeding experiment.

## Feeding experiment

According to the ambient field temperature, the feeding experiment was performed at three different temperatures: 29 °C, 33 °C, and 35 °C. The temperature of the sampling site was 29 °C, therefore this temperature treatment was used as the control. According to previous studies, a 6 °C temperature increase was observed in the waters of thermal discharge-affected areas in tropical zones (*Jiang et al., 2009*); therefore, 33 °C and 35 °C were used as the experimental temperatures, with the goal of investigating the rapid feeding response of copepods to acute heat shock.

Twenty liters of collected seawater were gently filtered through 0.45-μm polycarbonate membranes and then re-suspended in 5 L of FSW to concentrate. After thorough mixing, a 50 mL sample of concentrated seawater was removed and immediately fixed with Lugol's solution for cell counting as the prey concentration before the experiment. Another 200 mL seawater was taken and centrifuged at 3,000× g for 15 min, and the concentrated samples were pelleted by centrifugation at 12,000× g for 5 min. The cell pellets were re-suspended in 0.5 mL DNA buffer for DNA extraction. The concentrated seawater was then aliquoted into twelve 200-mL polycarbonate bottles to serve as a source of prey. For each treatment, a control group, without the addition of copepods, was prepared to monitor the growth of phytoplankton during the short exposure period. Each treatment had three replicates. Fifty adult individuals of *A. erythraea* were added to each experimental bottle and then incubated, at different temperatures, for 4 h in tanks with flowing seawater culture systems. After a 4 h incubation period, individual copepods were gently collected using 200 μm mesh and immediately fixed with neutral Lugol's solution to avoid defecation. Neutral Lugol's (no acetic acid added) was found to be effective in preserving DNA of phytoplankton and other copepod associated organisms (*Zhang & Lin, 2002*; *Hu et al., 2014*). There was no mortality of copepods during the short-term experiments. At the end of the experiments, 50 mL water samples were collected from each treatment and preserved with Lugol's solution as the prey concentration after the experiment. There was no measurable growth of phytoplankton during this period.

## Microscopic analysis of phytoplankton in seawater

Seawater samples were allowed to settle for 2 d in the dark, and then concentrated to 1 mL; they were then identified and enumerated in a Sedgwick-Rafter counting chamber under an Olympus BX51 microscope. The cell density of the seawater before and after the feeding experiment was obtained for the calculation of feeding parameters.

## Grazing rate and filter rate based on the phytoplankton cells

The grazing rate (GR, cells ind$^{-1}$h$^{-1}$) was calculated using the following equation:

$$\mathrm{GR} = V \times \frac{C_0 - C_t}{N \times t}.$$

The filter rate (FR, mL ind$^{-1}$h$^{-1}$) was calculated using the following equation:

$$FR = V \times \frac{lnC_0 - lnC_t}{N \times t}$$

where V (ml) is the volume of seawater in the experimental bottle, $N$ is the number of copepods in the bottle, and t (h) is the incubation time. $C_0$ and $C_t$ (cells mL$^{-1}$) are the prey concentrations (phytoplankton cells in the experimental seawater) before and after the feeding experiment, respectively.

## Molecular analysis of copepod diet

A previously reported procedure was followed (see *Hu et al., 2014*). Briefly, the sorted fixed copepod samples were thoroughly rinsed >3 times, with autoclaved 0.45-µm-filtered seawater and again with sterilized water several times. Copepods were examined under stereomicroscopy to ensure no attachment of other visible organisms on the body surface and then homogenized in a microfuge tube using a disposable micro pestle (Eppendorf, Hamburg, Germany). DNA from all copepod and seawater samples was extracted following the CTAB protocol. The DNA samples were PCR amplified using a PCR protocol based on a CEEC primer set (*Hu et al., 2014*). The PCR amplifications were concentrated using a Zymo DNA Clean & Concentrator TM-25 Kit. The target bands (∼0.8 kb) were recovered from a 1% agarose gel using Zymoclean$^{TM}$ Gel DNA Recovery Kit (Zymo Research, Irvine, CA, USA). The purified PCR products were ligated into a PMD18-T vector (Takara Bio Inc., Kusatsu, Japan) and transformed to DH-5$\alpha$ competent cells (Takara Bio Inc., Kusatsu, Japan). Approximately 50–80 clones for each copepod sample and 150–200 clones for each seawater sample were randomly picked and sequenced (Invitrogen sequencing company).

## Phylogenetic analysis of sequence data

Sequences obtained were trimmed of primer sequences and then searched against the GenBank database using the Basic Local Alignment Search Tool. The identified sequences were represented by the highest homologous sequences. Identified clones were categorized into plant-origin and animal-origin, and then a heat-map analysis was conducted using R (*R Core Team, 2017*). Our sequences were then aligned with the best hits using CLUSTAL W (1.8). The resulting alignment was imported into MEGA 6.0 to identify the best-fit nucleotide substitution model. The best-fit model Kimura 2, with gamma distribution (K2+G), was then employed for maximum likelihood (ML) analysis. The reliability of the tree topology was evaluated using bootstrap analysis with 1,000 replicates for ML analysis (*Hu et al., 2014*). Diversity indices (Chao-1, Simpson, and Shannon) were estimated using Past 3.05 to evaluate the sequencing depth of all the samples.

## Omnivory index and Ivlev's selectivity index

A percentage of each sequences from the whole clone library of each copepod sample was used to estimate the relative diet proportion. The trophic level and percentage were recorded to calculate the omnivory index (OI). This index was first introduced in the initial version of the Eco-path software and is calculated as the variance of the trophic level of a consumer's prey groups:

**Table 1  Grazing rate and filter rate of *Acartia erythraea* in different incubation treatments.**

| Treatment | Cell density at beginning (cells L$^{-1}$) | Cell density in the end (cells L$^{-1}$) | Copepod individuals | Incubation time | Filter rate (mL ind$^{-1}$h$^{-1}$) | Grazing rate (cells ind$^{-1}$h$^{-1}$) |
|---|---|---|---|---|---|---|
| 29 °C | $4.16 \times 10^4$ | $(1.56 \pm 0.07) \times 10^3$ | 50 | 4 h | $2.59 \pm 0.04$ | $40.04 \pm 0.07$ |
| 33 °C | $4.16 \times 10^4$ | $(2.46 \pm 0.11) \times 10^3$ | 50 | 4 h | $3.34 \pm 0.11$ | $40.14 \pm 0.11$ |
| 35 °C | $4.16 \times 10^4$ | $(1.34 \pm 0.14) \times 10^4$ | 50 | 4 h | $0.44 \pm 0.11$ | $28.20 \pm 0.14$ |

$$OI_i = \sum_{j=1}^{n}(TL_j - (TL_i - 1))^2 \cdot DC_{ij}$$

where $TL_j$ is the trophic level of prey $j$, $TL_i$ is the trophic level of the predator $i$, and, $DC_{ij}$ is the proportion prey $j$ constitutes to the diet of predator $i$. When the omnivory index value is zero, the consumer in question is specialized, i.e., it feeds on a single trophic level. A large value indicates that the consumer feeds on many trophic levels, or in other words is more omnivorous (*Pauly, Soriano-Bartz & Palomares, 1993*).

   Ivlev's selectivity index (*Ivlev, 1961*) was also used to determine the selectivity pattern of *A. erythraea* in the incubation experiment:

$$E_i = (r_i - P_i)/(r_i + P_i)$$

where $E_i$ is the Ivlev's selectivity index, $r_i$ is the relative abundance of prey $i$ in the diet of *A. erythraea* and $P_i$ is the relative abundance of the prey in ambient seawater. Observed values ranged from $-1$ to 1, where $-1$ indicates prey avoidance, 0 indicates that a prey species is being ingested at the same proportion as it is found in the environment, and 1 indicates a preference for a determined prey.

## RESULTS

### Grazing rate and filter rate of *A. erythraea* in different temperature treatments

Generally, the grazing rate and filter rate of *A. erythraea* showed an overall decrease with increasing temperature (Table 1). Compared with the control group (29 °C), the feeding of *A. erythraea* was increased slightly at 33 °C, with the grazing rate increasing from $40.04 \pm 0.07$ to $40.14 \pm 0.11$ cells ind$^{-1}$ h$^{-1}$ and the filter rate increasing from $2.59 \pm 0.04$ to $3.34 \pm 0.11$ mL ind$^{-1}$ h$^{-1}$, respectively. At 35 °C, the grazing rate of *A. erythraea* dramatically decreased to $28.20 \pm 0.14$ cells ind$^{-1}$ h$^{-1}$, and the filter rate decreased to $0.44 \pm 0.11$ mL ind$^{-1}$ h$^{-1}$. Increased temperature influenced the feeding behavior of *A. erythraea*; their ability to filter feed, especially phytoplankton, was limited when the temperature was higher than 33 °C.

### Dietary composition of *A. erythraea* from different temperature treatments

18S rDNA clone libraries were constructed for *A. erythraea* fed at different temperatures, as well as the plankton community of the concentrated field seawater used in the experiment.

**Table 2  Diversity indices of prey organisms in different *Acartia erythraea* samples.**

|  | 29 °C | 33 °C | 35 °C |
|---|---|---|---|
| Taxa | 12 | 10 | 10 |
| Individuals/clones | 50 | 50 | 50 |
| Simpson_1-D | 0.752 | 0.64 | 0.7832 |
| Shannon_H | 1.768 | 1.566 | 1.819 |
| Chao-1 | 19 | 10 | 11 |

Based on randomly sequenced clones, a total of 12, 10, and 10 taxa of prey were detected for *A. erythraea* at 29 °C, 33 °C, and 35 °C, respectively (Table 2). Although Chao-1 estimates indicated that the actual numbers of taxa were likely to be 19, 10, and 11, respectively, the prey taxa detected in the experimental groups (33 °C and 35 °C) were similar to the Chao-1 estimates, indicating an adequate coverage of diversity to demonstrate the effects of elevated temperature.

Despite the incomplete recovery of the entire dietary diversity in the control group (29 °C), prey taxa found in the samples consisted of a wide phylogenetic range of organisms, including diatoms, metazoans, protozoans, and fungi, as well as land plants (Fig. 1). In the 29 °C treatment, land plants (99% identical to *Hordeum jubatum* AF168852.1) and diatoms (98% identical to *Cylindrotheca closterium* LC054954.1) dominated the diet, accounting for 46% and 36%, respectively. Other numerically minor constituents such as fungi, protozoans, and Synurophyceae were also detected. In the 33 °C treatment, the same diatom species (*Cylindrotheca closterium*) was the most abundant prey taxa, comprising almost 58% of the diet. Other prey items belonged to the metazoans, protozoans, dinoflagellates, Synurophyceae, and fungi. The dietary spectrum of *A. erythraea* at 29 °C and 33 °C was similar. However, in the 35 °C treatment, a less diverse array of prey taxa was detected, with the contribution of metazoan taxa increasing up to 70%, but diatoms decreased to 16% (Fig. 1).

## Feeding preference changes with increasing temperature

Although the dietary spectrum was similar at different temperatures, the species numbers and relative proportions of each prey group (e.g., diatoms, metazoans, and protozoans) differed (Fig. 1). The omnivory index of *A. erythraea* at 29 °C and 33 °C was 0.1 and 0.24, respectively, while a remarkable increase (as high as 2.84) was observed in the 35 °C treatment. This indicates a more carnivorous feeding habit under temperature stress. The feeding habit of *A. erythraea* showed a likely shift from herbivory to carnivory when the temperature exceeded 33 °C. Furthermore, we divided all the prey items of *A. erythraea* from the different temperature treatments into three groups, metazoa, protozoa, and vege-prey (includes phytoplankton and plants), and calculated their Ivlev's selectivity index. *Acartia erythraea* showed positive feeding on vege-prey at 29 °C and 33 °C, but negative feeding on vege-prey at 35 °C. For metazoan prey, *A. erythraea* showed negative feeding at 29 °C but strong positive feeding in the 35 °C treatment, and the Ivlev's index increased from −0.62 to 0.75 (Fig. 2).

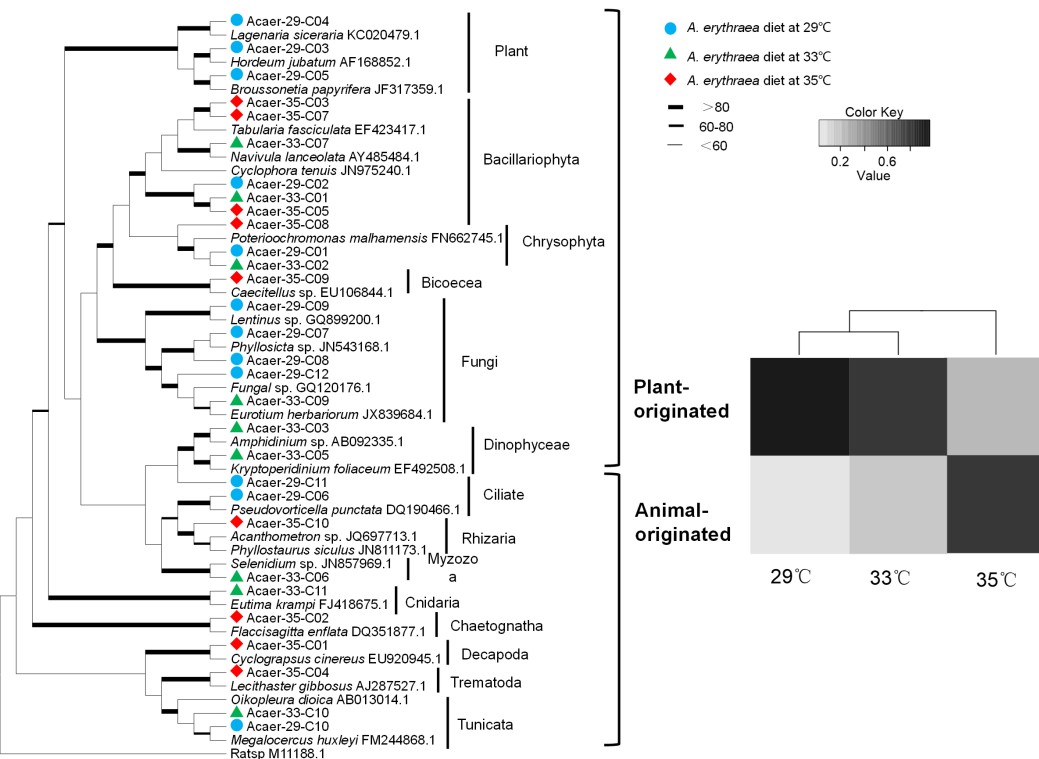

**Figure 1 Phylogenetic affiliations and taxonomic distribution of the prey organisms in the copepod *A. erythraea* in different temperature treatments.** The phylogenetic tree was established for prey taxa with the ML method and 1,000 bootstrap replicates, which was presented in the left part. The diet composition of *A. erythraea* from different temperatures was shown in the right part, and different colors represent the relative percentage of each prey group.

## DISCUSSION

Many studies have evaluated the environmental impact of elevated seawater temperatures near coastal power plants on marine organisms and coastal ecosystems (*Jiang et al., 2009*; *Li et al., 2011*; *Sommer & Lewandowska, 2011*; *Lewandowska et al., 2014a*). Most of these studies have focused on community structure or physiological responses in mesocosm incubations; however, little attention has been paid to trophic interactions, especially feeding responses to short-term thermal stress (*O'Connor et al., 2009*; *Saiz & Calbet, 2011*). Knowledge of the dietary composition of secondary consumers, such as copepods, exposed to warming environments is an important step in understanding exchanges of material and energy transfer through food webs affected by thermal effluents. Our results strongly suggest that increased temperature affected the active feeding behavior of *A. erythraea* when the temperature was higher than 33 °C; prey items showed reduced diversity and there was an obvious shift to carnivory during short-term exposure to high temperatures (35 °C).

As small omnivores, *Acartia* spp. usually dominate coastal zooplankton communities in food-sufficient seasons as they cannot adapt to low food concentrations. However, they

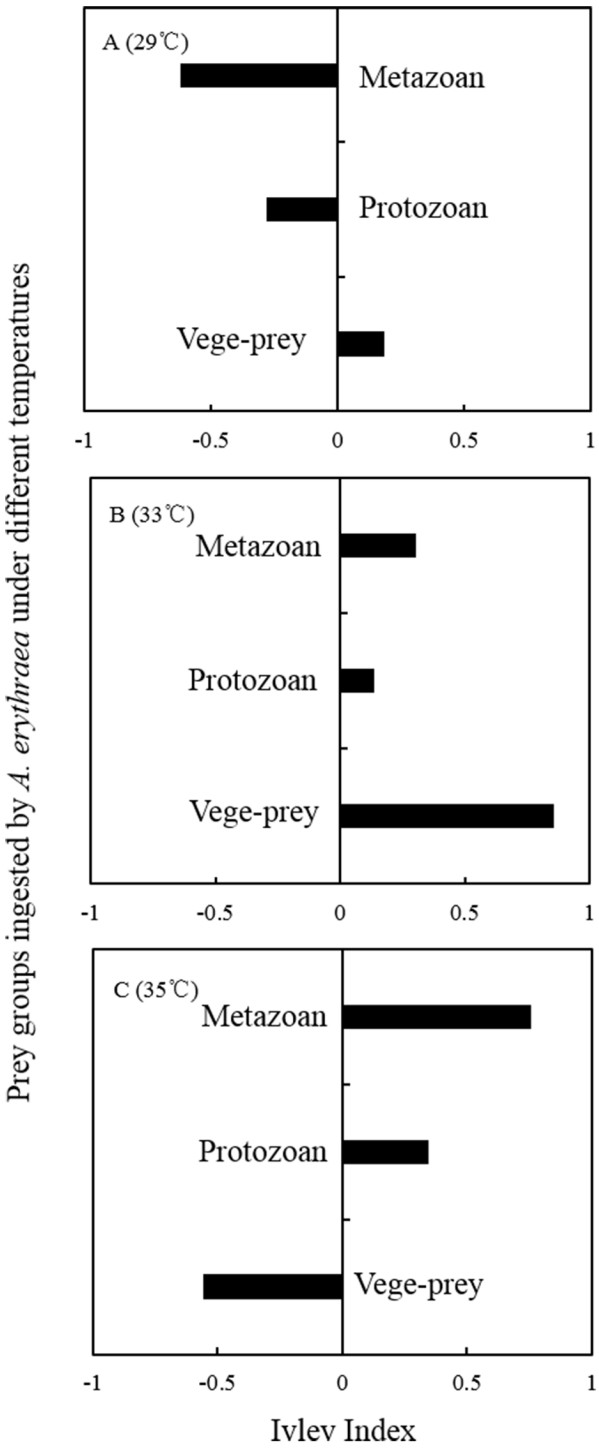

**Figure 2** **Ivlev Index of *A. erythraea* in different temperature treatments.** Vege-prey represents dietary organisms from phytoplankton and land plants.

can apply an opportunistic feeding strategy depending on prey concentration (*McKinnon, Duggan & De'ath, 2005*). Accordingly, the diet of *A. erythraea* in the present study was more diverse than that of other copepods in Sanya Bay, as identified by molecular detection; large amount of land-plant detritus were found in their diets (*Hu et al., 2015*). In the present study, a total of 25 taxa (genera or species) were represented in the diet, especially phytoplankton food. Diatoms were the most important prey group of *A. erythraea* and they dominated the dietary composition in the 29 °C and 33 °C treatments, with a small fraction of diatoms being consumed at 35 °C. These results confirmed that diatoms are an excellent food for copepods; as previously reported, they can produce large quantities of polyunsaturated fatty acids (PUFA), especially eicosapentaenoic acid (EPA). Some diatom species (e.g., *Thalassiosira weissflogii*) can support the entire growth phase of some *Acartia* spp. (e.g., *A. tonsa*) (*Dunstan et al., 1994*). *Cylindrotheca closterium* was the most abundant diatom species in the diet of *A. erythraea* in all the three treatments as it was the dominant phytoplankton species in the seawater and could be easily predated upon by copepods. The coastal coral waters of Sanya Bay are reportedly food-limited in terms of phytoplankton and *Hu et al. (2015)* found that copepods consumed large amounts of land plant detritus as a supplementary food source. This finding is consistent with our results from the control group, with land plants identified as the dominant food group. In comparison, diatoms were also abundant prey in the 29 °C and 33 °C treatments. The concentrated seawater we used for our incubation feeding experiments might have affected food availability, which was considered to be one of the major factors influencing copepod feeding rates in the field (*Saiz & Calbet, 2011*). In the present study, the concentration process increased the density of phytoplankton and might have decreased the energetic costs of copepod foraging on phytoplankton prey (*Alcaraz et al., 2014*). In the 35 °C treatment, only a small fraction of algal prey was identified; therefore, it is possible that elevated temperatures reduced copepod predation on phytoplankton.

Previous studies have shown that the optimum temperature for feeding of *Acartia* spp. was 25–30 °C, and grazing/filtering rates decreased beyond this range (*Milione & Zeng, 2008*). In the present study, we calculated the exact grazing/filtering rate of copepods based on phytoplankton cells, and found that the grazing rate of *A. erythraea* slightly increased in the 33 °C treatment, but decreased dramatically at 35 °C, indicating that the ability of copepods to actively feed might be weakened due to their reduced swimming ability at elevated temperatures (*Larsen, Madsen & Riisgård, 2008*). This is further supported by the observation that *A. erythraea* consumed prey with stronger swimming abilities, such as dinoflagellates and other protozoans, at 33 °C, whereas at 35 °C, it consumed more benthic diatom species (e.g., *Licmophora*, *Navicula*, *Cylindrotheca*). However, the most abundant prey group at elevated temperatures was metazoans (e.g., crustaceans, Chaetognatha). These prey might have originated from detritus-containing chaetognath or crustacean remnants, as chaetognaths and macrura larvae were abundant zooplankton groups in the sampling area, and have been reported to be prey for other copepods in Sanya Bay (*Yin et al., 2004*; *Hu et al., 2015*).

Except for diet diversity changes among the different treatments, the most obvious difference between the different temperatures was the shift from plant-originated food

items (i.e., phytoplankton and land plants) to animal prey (i.e., protozoans and metazoans). In the control treatment, plant-originated food items dominated the diet (>80%), while animal-derived foods only accounted for 4%. At 33 °C, although plant-originated food still comprised the majority of food items (∼75%), the proportion of protozoans increased to 10%. This finding is consistent with previous results that showed that zooplankton preferred to feed on microzooplankton during periods of increased temperature (*Lewandowska et al., 2014a*). On the other hand, the proportion of phytoplankton (mainly diatoms) in the diet decreased dramatically to 16% at 35 °C, while metazoan food items increased to almost 62%. Furthermore, the OI increased from 0.1 at 29 °C to 2.84 at 35 °C, indicating that *A. erythraea* turned to carnivory as a result of higher temperatures. These animal-based foods might be important at elevated temperatures as they have superior protein content quality compared to phytoplankton, and might be more easily assimilated by copepods as they have similar nutritional profiles (*Wang et al., 2014*). These results suggested that elevated temperature might narrow the food spectrum of copepods, but they are able to compensate by consuming more high energy prey items to sustain their metabolic needs (*Hammock et al., 2016*).

Previous studies have shown that zooplankton metabolic demands increased faster than their feeding rates in response to increased temperature, resulting in weaker interactions between zooplankton and phytoplankton and copepod top-down control over phytoplankton populations (*Rall et al., 2010*). In other words, energy available from phytoplankton as food items may be insufficient to support their metabolic needs; therefore, copepods might turn to consuming prey that can provide higher levels of energy. As reported by *Clarke, Holmes & Gore (1992)*, the energy content of lipids, proteins, and carbohydrates was 39.5 kJ g$^{-1}$, 23.9 kJ g$^{-1}$, and 17.5 kJ g$^{-1}$, respectively. Considering energy availability, metazoan prey (e.g., tunicates, chaetognaths, and crustacean amphipods) could provide copepods with more energy than phytoplankton (*Wang et al., 2014*). Additionally, the energy supplied by a metazoan diet with high levels of lipids and proteins, may be more accessible and easily utilized by copepods than plant food items, guaranteeing baseline metabolic demands during periods of stressful temperature conditions (*Colombo-Hixson et al., 2013*). This was an important feeding strategy for copepods in tropical seas near coastal power plants where abnormally high seawater temperatures, elevated up to 6–10 °C, were reported (*Poornima et al., 2006*).

Similar food availability in the incubation experiment masked, to some extent, the effects of temperature on feeding in the 33 °C treatment, as the diversity of the diet was slightly higher than that in the control group and the dominant prey group was the same as the control. Field data also showed that temperature might have a weaker effect on copepod feeding rates in natural waters because they would physiologically adapt to the feeding conditions (food availability) (*Saiz & Calbet, 2011*). Therefore, the feeding habit shift observed in the incubation experiment, which eliminated the influence of food availability and adaptation due to the short duration time, might reflect the exact effects of temperature on copepod feeding. The short-term response of tropical copepods found here was a little different from copepods in temperate seas, which prefer high-carbon-content phytoplankton food items at higher temperatures in natural seas (*Boersma et*

*al., 2016*). For tropical copepods, especially *Acartia*, optimizing energy acquisition for survival might be more important, similar to enhanced consumption of heterotrophic organisms in typical tropical habits (corals) under thermal stress (*Borell & Bischof, 2008*). Comprehensive studies should be conducted to clarify whether this short-term feeding response is a beneficial response for population-level thermal adaptation, or just guarantees immediate survival.

## CONCLUSIONS

Elevated temperatures affect the feeding behavior of *A. erythraea* by reducing their active feeding on phytoplankton and causing them to rely more on animal prey. Taken together, the findings of this study indicate that copepods seemed to adjust their feeding behavior to prioritize energy acquisition under short-term thermal stress. Although phytoplankton might be consumed by microzooplankton, previous studies have suggested an increased trophic pathway through protozooplankton under conditions of warming oceans as microzooplankton responded more quickly to the temperature changes. This change could reduce the efficiency of the upward trophic transfer of matter and energy flow through mesozooplankton under short term stress conditions, and thereby affect the efficiency of ecosystem cycling near coastal power plants. Due to the increasing need to generate electricity in coastal areas, coastal (fossil fuel or nuclear reactors) power plants must rely on seawater for cooling until renewable sources of electric production can be widely applied. Additionally, elevated seawater temperature caused by ongoing global warming will aggravate the impacts of thermal discharge on organisms near power plant outlets. Thus, our results indicate that rising seawater temperatures caused by coastal power plants can be of great significance in changing trophic pathways in regional seas. However, considering the variability in the responses of different species to thermal stress, further studies are needed on the feeding responses of tropical mesozooplankton at different trophic levels to precisely evaluate the effects of rising temperature on material and energy transfer through the food web.

## ACKNOWLEDGEMENTS

We are grateful to Cuilian Xu of University of Chinese Academy of Sciences for her help with sample collection and experimental setup.

### Funding

This study was partly supported by the Strategic Priority Research Program of Chinese Academy of Sciences (No. XDA13020100), the National Key Research and Development Project of China (No. 2016YFC0502800), and the Science and Technology Planning Projects of Guangdong province, China (No. 2015A020216013; 2017B030314052). There was no additional external funding received for this study. The funders had no role in study design, data collection and analysis, decision to publish, or preparation of the manuscript.

## Grant Disclosures

The following grant information was disclosed by the authors:

Strategic Priority Research Program of Chinese Academy of Sciences: XDA13020100.

National Key Research and Development Project of China: 2016YFC0502800.

Science and Technology Planning Projects of Guangdong province, China: 2015A020216013, 2017B030314052.

## Competing Interests

The authors declare there are no competing interests.

## Author Contributions

- Simin Hu conceived and designed the experiments, performed the experiments, analyzed the data, prepared figures and/or tables, authored or reviewed drafts of the paper, approved the final draft.
- Sheng Liu conceived and designed the experiments, contributed reagents/materials/-analysis tools, authored or reviewed drafts of the paper, approved the final draft.
- Lingli Wang analyzed the data, prepared figures and/or tables, approved the final draft.
- Tao Li performed the experiments, analyzed the data, approved the final draft.
- Hui Huang contributed reagents/materials/analysis tools, approved the final draft.

## DNA Deposition

The following information was supplied regarding the deposition of DNA sequences:

The partial 18S rDNA gene sequences obtained in this study are available at GenBank, accession numbers MH656588 to MH656617. The sequences are also available in a Supplemental File.

## Data Availability

The raw sequence data are provided in Data S1 and S2. These sequences were from clone libraries of sea water used in the feeding experiment and the copepod diet.

## Supplemental Information

Supplemental information for this article can be found online at http://dx.doi.org/10.7717/peerj.6129#supplemental-information.

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
