# Peer review of "Feeding response of the tropical copepod Acartia erythraea to short-term thermal stress: more animal-derived food was consumed"

_PeerJ, doi:10.7717/peerj.6129_

## Round 0.1 · original submission · Major Revisions

Both reviewers indicated the manuscript lacks certain types of information, both theoretical as well as methodological, and have made suggestions on manuscript organization. Also, some concerns were raised regarding the grammar and readability. Please pay particular attention to these and other comments in revising your manuscript.

·

Basic reporting

This submission is well written and adequately referenced. I do, however, recommend restructuring the article to make it more clear. For instance, it took me a while to understand where each calculation was coming from: I recommend more clearly separating the analysis of feeding on the basis of cell counts from those based on molecular work. This will not require a large re-write, just the insertion of some sub-headings.

Experimental design

The preparation procedures described subsequent to Line 149 is not very clear to me. Do I understand that the authors took four volumes of water and condensed that down on to a 0.47um filter to concentrate natural seston and plankton, then resuspended that material into a volume of filtered water equivalent to one of the original four volumes? If so, while the concentration procedure may not affect diatoms and detritus etc, it would gravely affect more delicate protozoans such as ciliates, ameboids and euglenoids, which would surely decompose once hitting the filter. By what factor is the particulate material in natural water concentrated?
Moreover, the authors introduce 50 Acartia erythraea into an experimental container of only 200mls, which corresponds to a concentration of 250,000 per cubic metre. It is not stated whether only adults were used in the experiment, but since the mesh size used in collection was 505um it is likely that only late copepodites and adults were available. In any case, this represents a massively inflated concentration of copepods from natural field densities (which were?) and could have implications for the interpretation of the results.
Otherwise, the protocols adopted for the experiments were perfectly standard.
At line 159 it is stated that a 50ml water sample was collected and preserved with Lugols, but there is no information on how the plankton were enumerated.
I was surprised to read that copepods preserved in Lugols were used for molecular analysis – I would have thought that the iodine would have compromised the extraction of nucleic acids. I’d like to see this clarified in the text – though I accept that this paper seems to be part of a body of work by the senior author in which these issues may have been addressed. That said, I’d say this contribution ought to stand on its own.

At line 228 the authors introduce a parameter called the Chao-1 index. I hadn’t heard of this before, and all the diversity indices used ought to be described in the Methods section.

Validity of the findings

I was attracted to review this manuscript because I found the principal finding – the trend toward carnivory in warmer temperatures – to be very interesting in the broader context of a warming world. Just how much the effluent from power stations affects local plankton communities depends on the oceanography of the discharge area. If tides and currents are appreciable, I would think that such effects would be extremely localized.

That said, the findings are of general interest even though they are placed in the context of local effects, and the molecular analyses of prey taxa are broadly applicable and very welcome.

Reviewer 2 ·

Basic reporting

Overall, this manuscript does provide a clear hypothesis and description of how it was tested, but lacks a distinct amount of detail in key sections of the manuscript.

For example, the introduction lacks a sense of depth and clarity as to the importance of short-term increases in water temperatures on the physiology of copepods in particular. Changes in water temperature ay affect copepod physiological processes directly, by altering metabolic rates and energetic demands, or indirectly through alterations in the availability of prey. On Lines 85-87 the authors state “Elevation of temperature within a moderate range could promote the growth and metabolic rates of copepod, but if the temperatures exceed this range, the physiological state of copepods might be changed.” However, this statement needs to be expanded upon, and there needs to be a more explicit link drawn between increases in water temperature and the specific physiological changes that may occur, how they may occur, and why. For example, why are the authors looking specifically at feeding responses? Is this due to the presumed direct effects of temperature on the physiology of copepods, or indirectly through dietary switching as suggested in Lines 93-103. While both of these are clearly important processes, each should be explained in more detail, and a clearer separation of the two is necessary.

Furthermore, more context about thermal effluent and its effects on surrounding environments is needed. How prevalent are coastal power plants in tropical areas? How often does thermal discharge occur, and how drastically is the water temperature increased? How fast does this change occur, and how long does the water stay elevated? These are important details that will help inform the reader not only the magnitude of the problem at hand, but also about the validity of the experimental design. This could be emphasized in the discussion as well, since it was highlighted at the forefront of the manuscript. What are the implications of potential dietary shifts, given the fact that most organisms are facing not only increased temperatures due to global climate change, but also due to anthropogenic impacts?

Lastly, while Figure 2 is clear and demonstrates the authors’ data well, Figure 1 is slightly difficult to interpret, and somewhat cumbersome.

Experimental design

Generally speaking, details are missing from the description of the experimental design that may be important to the reader. For example, the authors state on Lines 132-133 that mature females were separated out from the samples, but then do not state if they tested females, males, or a combination of both.

It is also unclear if the authors ensured equal concentration of prey cells in each treatment, as the concentration of prey could influence feeding rate as well. While this did not bias results, since the authors calculated grazing rate using the initial cell count of each vial individually, the density of prey could influence copepod behavior in the absence of a temperature effect. If this was indeed the case, and I have misinterpreted the authors, I apologize, but perhaps this is an indication that it should be made clearer.

Similarly, the authors report Simpson, Shannon, and Chao indices, yet make no mention of these in the methods section.

Validity of the findings

Generally, the interpretation of the data by the authors is scientifically sound and well supported. Indeed, the authors provide great detail for their interpretations of the results, and are clear in the purported mechanisms that may have contributed to what was quantified. Overall, I found this section of the paper most pleasing to read.

Additional comments

While the writing does not inhibit the understanding of the reader, this manuscript should be reviewed and edited substantially by a native English speaker to correct many of the grammatical and structural errors of the text. This would improve the flow of the text, and might help mediate any confusion on the readers’ part.

---

## Round 0.2 · Minor Revisions

Thank you for your efforts in revising the manuscript which is now scientifically acceptable.

#

---

## Round 0.3 · accepted · Accept

Thank you for your efforts in revising your manuscript and making it more readable.

#